# Tactical Optimism and Pessimism for Deep Reinforcement Learning

**Ted Moskovitz**
Gatsby Unit, UCL
ted@gatsby.ucl.ac.uk

**Jack Parker-Holder**
University of Oxford
jackph@robots.ox.ac.uk

**Aldo Pacchiano**
Microsoft Research
apacchiano@microsoft.com

**Michael Arbel**
Université Grenoble Alpes, Inria, CNRS*
michael.n.arbel@gmail.com

**Michael I. Jordan**
University of California, Berkeley
jordan@cs.berkeley.edu

## Abstract

In recent years, deep off-policy actor-critic algorithms have become a dominant approach to reinforcement learning for continuous control. One of the primary drivers of this improved performance is the use of pessimistic value updates to address function approximation errors, which previously led to disappointing performance. However, a direct consequence of pessimism is reduced exploration, running counter to theoretical support for the efficacy of optimism in the face of uncertainty. So which approach is best? In this work, we show that the most effective degree of optimism can vary both across tasks and over the course of learning. Inspired by this insight, we introduce a novel deep actor-critic framework, *Tactical Optimistic and Pessimistic* (TOP) estimation, which switches between optimistic and pessimistic value learning *online*. This is achieved by formulating the selection as a multi-arm bandit problem. We show in a series of continuous control tasks that TOP outperforms existing methods which rely on a fixed degree of optimism, setting a new state of the art in challenging pixel-based environments. Since our changes are simple to implement, we believe these insights can easily be incorporated into a multitude of off-policy algorithms.

## 1 Introduction

Reinforcement learning (RL) has begun to show significant empirical success in recent years, with value function approximation via deep neural networks playing a fundamental role in this success [37, 49, 5]. However, this success has been achieved in a relatively narrow set of problem domains, and an emerging set of challenges arises when one considers placing RL systems in larger systems. In particular, the use of function approximators can lead to a positive bias in value computation [53], and therefore systems that surround the learner do not receive an honest assessment of that value. One can attempt to turn this vice into a virtue, by appealing to a general form of the optimism-under-uncertainty principle—overestimation of the expected reward can trigger exploration of states and actions that would otherwise not be explored. Such exploration can be dangerous, however, if there is not a clear understanding of the nature of the overestimation.

---

*Work mostly completed at the Gatsby Unit.

35th Conference on Neural Information Processing Systems (NeurIPS 2021).

This tension has not been resolved in the recent literature on RL approaches to continuous-control problems. On the one hand, some authors seek to correct the overestimation, for example by using the minimum of two value estimates as a form of approximate lower bound [20]. This approach can be seen as a form of *pessimism* with respect to the current value function. On the other hand, [14] have argued that the inherent optimism of approximate value estimates is actually useful for encouraging exploration of the environment and/or action space. Interestingly, both sides have used their respective positions to derive state-of-the-art algorithms. How can this be, if their views are seemingly opposed? Our key hypothesis is the following:

*The degree of estimation bias, and subsequent efficacy of an optimistic strategy, varies as a function of the environment, the stage of optimization, and the overall context in which a learner is embedded.*

This hypothesis motivates us to view optimism/pessimism as a spectrum and to investigate procedures that actively move along that spectrum during the learning process. We operationalize this idea by measuring two forms of uncertainty that arise during learning: *aleatoric uncertainty* and *epistemic uncertainty*. These notions of uncertainty, and their measurement, are discussed in detail in Section 5.1. We then further aim to control the effects of these two kinds of uncertainty, making the following learning-theoretic assertion:

*When the level of bias is unknown, an adaptive strategy can be highly effective.*

In this work, we investigate these hypotheses via the development of a new framework for value estimation in deep RL that we refer to as *Tactical Optimism and Pessimism* (TOP). This approach acknowledges the inherent uncertainty in the level of estimation bias present, and rather than adopt a blanket optimistic or pessimistic strategy, it estimates the optimal approach *on the fly*, by formulating the optimism/pessimism dilemma as a multi-armed bandit problem. Furthermore, TOP explicitly isolates the aleatoric and epistemic uncertainty by representing the environmental return using a distributional critic and model uncertainty with an ensemble. The overall concept is summarized in Figure 1.

We show in a series of experiments that not only does the efficacy of optimism indeed vary as we suggest, but TOP is able to capture the best of both worlds, achieving a new state of the art for challenging continuous control problems.

Our main contributions are as follows:

- Our work shows that the efficacy of optimism for a fixed function approximator varies across environments and during training for reinforcement learning with function approximation.

- We propose a novel framework for value estimation, *Tactical Optimism and Pessimism* (TOP), which learns to balance optimistic and pessimistic value estimation online. TOP frames the choice of the degree of optimism or pessimism as a multi-armed bandit problem.

- Our experiments demonstrate that these insights, which require only simple changes to popular algorithms, lead to state-of-the-art results on both state- and pixel-based control.

## 2 Related Work

Much of the recent success of off-policy actor-critic algorithms build on DDPG [34], which extended the deterministic policy gradient [49] approach to off-policy learning with deep networks, using insights from DQN [37]. Like D4PG [8], we combine DPG with distributional value estimation. However, unlike D4PG, we use two critics, a quantile representation rather than a categorical distribution [10], and, critically, we actively manage the tradeoff between optimism and pessimism. We also note several other success stories in the actor-critic vein, including TD3, SAC, DrQ, and PI-SAC

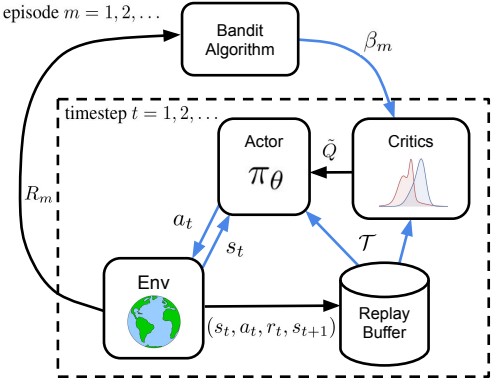

Figure 1: Visualization of the TOP framework. Blue arrows denote stochastic variables.

[20, 21, 58, 33]; these represent the state-of-the-
art for continuous control and will serve as a baseline for our experiments.

The principle of optimism in the face of uncertainty [3, 30, 59] provides a design tool for algorithms that trade off exploitation (maximization of the reward) against the need to explore state-action pairs with high epistemic uncertainty. The theoretical tool for evaluating the success of such designs is the notion of regret, which captures the loss incurred by failing to explore. Regret bounds have long been used in research on multi-armed bandits, and they have begun to become more prominent in RL as well, both in the tabular setting [26, 18, 19, 4, 9, 55], and in the setting of function approximation [28, 57]. However, optimistic approaches have had limited empirical success when combined with deep neural networks in RL [14]. To be successful, these approaches need to be optimistic enough to upper bound the true value function while maintaining low estimation error [41]. This becomes challenging when using function approximation, and the result is often an uncontrolled, undesirable overestimation bias.

Recently, there has been increasing evidence in support of the efficacy of adaptive algorithms [6, 48, 46, 45]. An example is Agent57 [5], the first agent to outperform the human baseline for all 57 games in the Arcade Learning Environment [11]. Agent57 adaptively switches among different exploration strategies. Our approach differs in that it aims to achieve a similar goal by actively varying the level of optimism in its value estimates.

Finally, our work is also related to automated RL (AutoRL), as we can consider TOP to be an example of an on-the-fly learning procedure [15, 39, 29]. An exciting area of future work will be to consider the interplay between the degree of optimism and model hyperparameters such as architecture and learning rate, and whether they can be adapted simultaneously.

## 3 Preliminaries

Reinforcement learning considers the problem of training an agent to interact with its environment so as to maximize its cumulative reward. Typically, a task and environment are cast as a Markov decision process (MDP), formally defined as a tuple $(\mathcal{S}, \mathcal{A}, p, r, \gamma)$, where $\mathcal{S}$ is the state space, $\mathcal{A}$ is the space of possible actions, $p : \mathcal{S} \times \mathcal{A} \to \mathcal{P}(\mathcal{S})$ is a transition kernel, $r : \mathcal{S} \times \mathcal{A} \to \mathbb{R}$ is the reward function, and $\gamma \in [0, 1)$ is a discounting factor. For a given policy $\pi$, the *return* $Z^\pi = \sum_t \gamma^t r_t$, is a random variable representing the sum of discounted rewards observed along one trajectory of states obtained from following $\pi$ until some time horizon $T$, potentially infinite. Given a parameterization of the set of policies, $\{\pi_\theta : \theta \in \Theta\}$, the goal is to update $\theta$ so as to maximize the *expected return*, or discounted cumulative reward, $J(\theta) = \mathbb{E}_\pi \left[ \sum_t \gamma^t r_t \right] = \mathbb{E}[Z^\pi]$.

Actor-critic algorithms are a framework for solving this problem in which the policy $\pi$, here known as the *actor*, is trained to maximize expected return, while making use of a *critic* that evaluates the actions of the policy. Typically, the critic takes the form of a value function which predicts the expected return under the current policy, $Q^\pi(s, a) := \mathbb{E}_\pi[Z_t | s_t = s, a_t = a]$. When the state space is large, $Q^\pi$ may be parameterized by a model with parameters $\phi$. The *deterministic policy gradient* (DPG) theorem [50] shows that gradient ascent on $J$ can be performed via

$$\nabla_\theta J(\theta) = \mathbb{E}_\pi[\nabla_a Q^\pi(s, a)|_{a=\pi(s)} \nabla_\theta \pi_\theta(s)]. \tag{1}$$

The critic is updated separately, usually via SARSA [51], which, given a transition $s_t, a_t \to r_{t+1}, s_{t+1}$, forms a learning signal via semi-gradient descent on the squared temporal difference (TD) error, $\delta_t^2$, where

$$\delta_t := y_t - Q^\pi(s_t, a_t) = r_{t+1} + \gamma Q^\pi(s_{t+1}, \pi(s_{t+1})) - Q^\pi(s_t, a_t), \tag{2}$$

and where $y_t$ is the *Bellman target*. Rather than simply predicting the mean of the return $Z^\pi$ under the current policy, it can be advantageous to learn a full *distribution* of $Z^\pi$ given the current state and action, $\mathcal{Z}^\pi(s_t, a_t)$ [10, 17, 16, 47]. In this framework, the return distribution is typically parameterized via a set of $K$ functionals of the distribution (e.g., quantiles or expectiles) which are learned via minimization of an appropriate loss function. For example, the $k$th quantile of the distribution at state $s$ and associated with action $a$, $q_k(s, a)$, can be learned via gradient descent on the *Huber loss* [25] of the *distributional Bellman error*, $\delta_k = \hat{Z} - q_k(s, a)$, for $\hat{Z} \sim \mathcal{Z}^\pi(\cdot | s, a)$. While $\hat{Z}$ is formally defined as a sample from the return distribution, $\delta_k$ is typically computed in practice as $K^{-1} \sum_{j=1}^K r + \gamma q_j(s, a) - q_k(s, a)$ [17].

## 4 Optimism versus Pessimism

**Reducing overestimation bias with pessimism**   It was observed by [53] that Q-learning [56] with function approximation is biased towards overestimation. Noting that this overestimation bias can introduce instability in training, [20] introduced the *Twin Delayed Deep Deterministic* (TD3) policy gradient algorithm to correct for the bias. TD3 can be viewed as a pessimistic heuristic in which values are estimated via a SARSA-like variant of double Q-learning [24] and the Bellman target is constructed by taking the minimum of *two* critics:

$$y_t = r_{t+1} + \gamma \min_{i \in \{1,2\}} Q^{\pi}_{\phi_i}(s, \pi_\theta(s) + \epsilon). \tag{3}$$

Here $\epsilon \sim \text{clip}(\mathcal{N}(0, s^2), -c, c)$ is drawn from a clipped Gaussian distribution ($c$ is a constant). This added noise is used for smoothing in order to prevent the actor from overfitting to narrow peaks in the value function. Secondly, TD3 *delays* policy updates, updating value estimates several times between each policy gradient step. By taking the minimum of two separate critics and increasing the number of critic updates for each policy update, this approach takes a pessimistic view on the policy's value in order to reduce overestimation bias. These ideas have become ubiquitous in state-of-the-art continuous control algorithms [7], such as SAC, RAD, (PI)-SAC [21, 31, 33].

**Optimism in the face of uncertainty**   While it is valuable to attempt to correct for overestimation of the value function, it is also important to recall that overestimation can be viewed as a form of optimism, and as such can provide a guide for exploration, a necessary ingredient in theoretical treatments of RL in terms of regret [27, 26, 4]. In essence, the effect of optimistic value estimation is to induce the agent to explore regions of the state space with high epistemic uncertainty, encouraging further data collection in unexplored regions. Moreover, [14] found that reducing value estimates, as done in pessimistic algorithms, can lead to *pessimistic underexploration*, in which actions that could lead to experience that gives the agent a better long-term reward. To address this problem, [14] introduced the *Optimistic Actor-Critic* (OAC) algorithm, which trains an *exploration policy* using an optimistic upper bound on the value function while constructing targets for learning using the lower bound of [20]. OAC demonstrated improved performance compared to SAC, hinting at a complex interplay between optimism and pessimism in deep RL algorithms.

**Trading off optimism and pessimism**   As we have discussed, there are arguments for both optimism and pessimism in RL. Optimism can aid exploration, but if there is significant estimation error, then a more pessimistic approach may be needed to stabilize learning. Moreover, both approaches have led to algorithms that are supported by strong empirical evidence. We aim to reconcile these seemingly contradictory perspectives by hypothesizing that the relative contributions of these two ingredients can vary depending on the nature of the task, with relatively simple settings revealing predominantly one aspect. As an illustrative example, we trained "Optimistic" and "Pessimistic" versions of the same deep actor-critic algorithm (details in Section 6) for two different tasks and compared their performance in Figure 2. As we can see, in the HalfCheetah task, the Optimistic agent outperforms the Pessimistic agent, while in the Hopper task, the opposite is true. This result suggests that the overall phenomenon is multi-faceted and active management of the overall optimism-pessimism trade-off is necessary. Accordingly, in the current paper we propose the use of an adaptive approach in which the degree of optimism or pessimism is adjusted dynamically during training. As a consequence of this approach, the optimal degree of optimism can vary across tasks and over the course of a single training run as the model improves. Not only does this approach reconcile the seemingly contradictory perspectives in the literature, but it also can outperform each individual framework in a wider range of tasks.

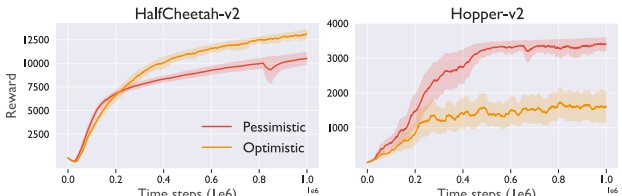

Figure 2: Optimistic and Pessimistic algorithms averaged over 10 seeds. Shading is one half std.

## 5 Tactical Optimistic and Pessimistic Value Estimation

TOP is based on the idea of adaptive optimism in the face of uncertainty. We begin by discussing how TOP represents uncertainty and then turn to a description of the mechanism by which TOP dynamically adapts during learning.

## 5.1 Representing uncertainty in TOP estimation

TOP distinguishes between two types of uncertainty—*aleatoric uncertainty* and *epistemic uncertainty*—and represents them using two separate mechanisms.

**Aleatoric uncertainty** reflects the noise that is inherent to the environment regardless of the agent's understanding of the task. Following [10, 17, 16, 47], TOP represents this uncertainty by learning the full return distribution, $\mathcal{Z}^\pi(s,a)$, for a given policy $\pi$ and state-action pair $(s,a)$ rather than only the expected return, $Q^\pi(s,a) = \mathbb{E}[Z^\pi(s,a)]$, $Z^\pi(s,a) \sim \mathcal{Z}^\pi(\cdot|s,a)$. Note here we are using $Z^\pi(s,a)$ to denote the return random variable and $\mathcal{Z}^\pi(s,a)$ to denote the return *distribution* of policy $\pi$ for state-action pair $(s,a)$. Depending on the stochasticity of the environment, the distribution $\mathcal{Z}^\pi(s,a)$ is more or less spread out, thereby acting as a measure of aleatoric uncertainty.

**Epistemic uncertainty** reflects lack of knowledge about the environment and is expected to decrease as the agent gains experience. TOP uses this uncertainty to quantify how much an optimistic belief about the return differs from a pessimistic one. Following [14], we model epistemic uncertainty via a Gaussian distribution with mean $\bar{Z}(s,a)$ and standard deviation $\sigma(s,a)$ of the quantile estimates across multiple critics as follows:

$$Z^\pi(s,a) \stackrel{d}{=} \bar{Z}(s,a) + \epsilon\sigma(s,a), \tag{4}$$

where $\stackrel{d}{=}$ indicates equality in distribution. However, unlike in [14], where the parameters of the Gaussian are deterministic, we treat both $\bar{Z}(s,a)$ and $\sigma(s,a)$ as random variables underlying a Bayesian representation of aleatoric uncertainty. As we describe next, only $\mathcal{Z}^\pi(s,a)$ is modeled (via a quantile representation), hence $\bar{Z}(s,a)$ and $\sigma(s,a)$ are unknown. Proposition 1 shows how to recover them from $Z^\pi(s,a)$ and is proven in Appendix E.

**Proposition 1.** *The quantile function $q_{\bar{Z}(s,a)}$ for the distribution of $\bar{Z}$ is given by:*

$$q_{\bar{Z}(s,a)} = \mathbb{E}_\epsilon\left[q_{Z^\pi(s,a)}\right], \tag{5}$$

*where $q_{Z^\pi(s,a)}$ is the quantile function of $\mathcal{Z}^\pi(s,a)$ knowing $\epsilon$ and $\sigma(s,a)$ and $\mathbb{E}_\epsilon$ denotes the expectation w.r.t. $\epsilon \sim \mathcal{N}(0,1)$. Moreover, $\sigma^2(s,a)$ satisfies:*

$$\sigma^2(s,a) = \mathbb{E}_\epsilon[\|\bar{Z}(s,a) - Z^\pi\|^2]. \tag{6}$$

**Quantile approximation**   Following [17], TOP represents the return distribution $Z^\pi(s,a)$ using a *quantile approximation*, meaning that it forms $K$ statistics, $q^{(k)}(s,a)$, to serve as an approximation of the quantiles of $\mathcal{Z}^\pi(s,a)$. The quantiles $q^{(k)}(s,a)$ can be learned as the outputs of a parametric function—in our case, a deep neural network—with parameter vector $\phi$. To measure epistemic uncertainty, TOP stores two estimates, $\mathcal{Z}_1^\pi(s,a)$ and $\mathcal{Z}_2^\pi(s,a)$, with respective quantile functions $q_1^{(k)}(s,a)$ and $q_2^{(k)}(s,a)$ and parameters $\phi_1$ and $\phi_2$. This representation allows for straightforward estimation of the mean $\bar{Z}(s,a)$ and variance $\sigma(s,a)$ in (4) using Proposition 1. Indeed, applying (5) and (6) and treating $Z_1^\pi(s,a)$ and $Z_2^\pi(s,a)$ as exchangeable draws from (4), we approximate the quantiles $q_{\bar{Z}(s,a)}$ and $q_{\sigma(s,a)}$ of the distribution of $\bar{Z}(s,a)$ and $\sigma(s,a)$ as follows:

$$\bar{q}^{(k)}(s,a) = \frac{1}{2}\left(q_1^{(k)}(s,a) + q_2^{(k)}(s,a)\right), \qquad \sigma^{(k)}(s,a) = \sqrt{\sum_{i=1}^2 \left(q_i^{(k)}(s,a) - \bar{q}^{(k)}(s,a)\right)^2}. \tag{7}$$

Next, we will show these approximations can be used to define an exploration strategy for the agent.

## 5.2 An uncertainty-based strategy for exploration

We use the quantile estimates defined in (7) to construct a *belief distribution* $\tilde{\mathcal{Z}}^\pi(s,a)$ over the expected return whose quantiles are defined by

$$q_{\tilde{\mathcal{Z}}^\pi(s,a)} = q_{\bar{Z}(s,a)} + \beta q_{\sigma(s,a)}. \tag{8}$$

This belief distribution $\tilde{\mathcal{Z}}^\pi(s,a)$ is said be *optimistic* when $\beta \geq 0$ and *pessimistic* when $\beta < 0$. The amplitude of optimism or pessimism is measured by $\sigma(s,a)$, which quantifies epistemic uncertainty.

The degree of optimism depends on $\beta$ and is adjusted dynamically during training, as we will see in Section 5.3. Note that $\beta$ replaces $\epsilon \sim \mathcal{N}(0,1)$, making the belief distribution non-Gaussian.

**Learning the critics.** TOP uses the belief distribution in (8) to form a target for both estimates of the distribution, $\mathcal{Z}_1^\pi(s,a)$ and $\mathcal{Z}_2^\pi(s,a)$. To achieve this, TOP computes an approximation of $\tilde{\mathcal{Z}}^\pi(s,a)$ using $K$ quantiles $\tilde{q}^{(k)} = \bar{q}^k + \beta\sigma^{(k)}$. The temporal difference error for each $\mathcal{Z}_i^\pi(s,a)$ is given by $\delta_i^{(j,k)} := r + \gamma\tilde{q}^{(j)} - q_i^{(k)}$ with $i \in \{1,2\}$ and where $(j,k)$ ranges over all possible combinations of quantiles. Finally, following the quantile regression approach in [17], we minimize the Huber loss $\mathcal{L}_{\text{Huber}}$ evaluated at each distributional error $\delta_i^{(j,k)}$, which provides a gradient signal to learn the distributional critics as given by (9):

$$\Delta\phi_i \propto \sum_{1 \le k,j \le K} \nabla_{\phi_i}\mathcal{L}_{\text{Huber}}(\delta_i^{(j,k)}). \tag{9}$$

The overall process is summarized in Algorithm 2.

**Learning the actor.** The actor is trained to maximize the expected value $\tilde{Q}(s,a)$ under the belief distribution $\tilde{\mathcal{Z}}^\pi(s,a)$. Using the quantile approximation, $\tilde{Q}(s,a)$ is simply given as an average over $\tilde{q}^{(k)}$: $\tilde{Q}(s,a) = \frac{1}{K}\sum_{k=1}^K \tilde{q}^{(k)}(s,a)$. The update of the actor follows via the DPG gradient:

$$\Delta\theta \propto \nabla_a\tilde{Q}(s,a)|_{a=\pi_\theta(s)}\nabla_\theta\pi_\theta(s). \tag{10}$$

This process is summarized in Algorithm 3. To reduce variance and leverage past experience, the critic and actor updates in (9) and (10) are both averaged over $N$ transitions, $(s,a,r,s')_{n=1}^N$, sampled from a replay buffer $\mathcal{B}$ [35].

In the special case of $\beta = -1/\sqrt{2}$, the average of (8) reduces to $\min_i Z_i^\pi(s,a)$ and (10) recovers a distributional version of TD3, a pessimistic algorithm. On the other hand, when $\beta \ge 0$, the learning target is optimistic with respect to the current value estimates, recovering a procedure that can be viewed as a distributional version of the optimistic algorithm of [14]. However, in our case, when $\beta \ge 0$ the learning target is also optimistic. Hence, (9) and (10) can be seen as a generalization of the existing literature to a distributional framework that can recover both optimistic and pessimistic value estimation depending on the sign of $\beta$. In the next section we propose a principled way to adapt $\beta$ during training to benefit from both the pessimistic and optimistic facets of our approach.

### 5.3 Optimism and pessimism as a multi-arm bandit problem

As we have seen (see Figure 2), the optimal degree of optimism or pessimism for a given algorithm may vary across environments. As we shall see, it can also be beneficial to be more or less optimistic over the course of a single training run. It is therefore sensible for an agent to adapt its degree of optimism dynamically in response to feedback from the environment. In our framework, the problem can be cast in terms of the choice of $\beta$. Note that the evaluation of the effect of $\beta$ is a form of bandit feedback, where learning episodes tell us about the absolute level of performance associated with a particular value of $\beta$, but do not tell us about relative levels. We accordingly frame the problem as a multi-armed bandit problem, using the Exponentially Weighted Average Forecasting algorithm [13]. In our setting, each bandit arm represents a particular value of $\beta$, and we consider $D$ experts making recommendations from a discrete set of values $\{\beta_d\}_{d=1}^D$. After sampling a decision $d_m \in \{1,\ldots,D\}$ at episode $m$, we form a distribution $\mathbf{p}_m \in \Delta_D$ of the form $\mathbf{p}_m(d) \propto \exp(w_m(d))$. The learner receives a feedback signal, $f_m \in \mathbb{R}$, based on this choice. The parameter $w_m$ is updated as follows:

$$w_{m+1}(d) = \begin{cases} w_m(d) + \eta\frac{f_m}{\mathbf{p}_m(d)} & \text{if } d = d_m \\ w_m(d) & \text{otherwise,} \end{cases} \tag{11}$$

for a step size parameter $\eta > 0$. Intuitively, if the feedback signal obtained is high and the current probability of selecting a given arm is low, the likelihood of selecting that arm again will increase. For the feedback signal $f_m$, we use *improvement in performance*. Concretely, we set $f_m = R_m - R_{m-1}$, where $R_m$ is the cumulative reward obtained in episode $m$. Henceforth, we denote by $\mathbf{p}_m^\beta$ the exponential weights distribution over $\beta$ values at episode $m$.

Our approach can be thought of as implementing a form of model selection similar to that of [44], where instead of maintaining distinct critics for each optimism choice, we simply update the same pair of critics using the choice of $\beta$ proposed by the bandit algorithm. For a more thorough discussion of TOP's connection to model selection, see Appendix D.

---

| Algorithm 1: TOP-TD3 |
|---|

1: Initialize critic networks $Q_{\phi_1}$, $Q_{\phi_2}$ and actor $\pi_\theta$
   Initialize target networks $\phi_1' \leftarrow \phi_1$, $\phi_2' \leftarrow \phi_2$, $\theta' \leftarrow \theta$
   Initialize replay buffer and bandit probabilities $\mathcal{B} \leftarrow \emptyset$, $\mathbf{p}_1^\beta \leftarrow \mathcal{U}([0,1]^D)$
2: **for** episode in $m = 1, 2, \ldots$ **do**
3:     Initialize episode reward $R_m \leftarrow 0$
4:     Sample optimism $\beta_m \sim \mathbf{p}_m^\beta$
5:     **for** time step $t = 1, 2, \ldots, T$ **do**
6:         Select noisy action $a_t = \pi_\theta(s_t) + \epsilon$, $\epsilon \sim \mathcal{N}(0, s^2)$, obtain $r_{t+1}, s_{t+1}$
7:         Add to total reward $R_m \leftarrow R_m + r_{t+1}$
8:         Store transition $\mathcal{B} \leftarrow \mathcal{B} \cup \{(s_t, a_t, r_{t+1}, s_{t+1})\}$
9:         Sample $N$ transitions $\mathcal{T} = (s, a, r, s')_{n=1}^N \sim \mathcal{B}$.
10:        UpdateCritics($\mathcal{T}, \beta_m, \theta', \phi_1', \phi_2'$)
11:        **if** $t \mod b$ **then**
12:            UpdateActor($\mathcal{T}, \beta_m, \theta, \phi_1, \phi_2$)
13:            Update $\phi_i'$: $\phi_i' \leftarrow \tau\phi_i + (1-\tau)\phi_i'$, $i \in \{1, 2\}$
14:            Update $\theta'$: $\theta' \leftarrow \tau\theta + (1-\tau)\theta'$
15:        **end for**
16:        Update bandit $\mathbf{p}^\beta$ weights using (11)
17: **end for**

---

### 5.4 The TOP framework

The general TOP framework can be applied to any off-policy actor-critic architecture. As an example, an integration of the procedure with TD3 (TOP-TD3) is shown in Algorithm 1, with key differences from TD3 highlighted in purple. Like TD3, we apply target networks, which use slow-varying averages of the current parameters, $\theta, \phi_1, \phi_2$, to provide stable updates for the critic functions. The target parameters $\theta', \phi_1', \phi_2'$ are updated every $b$ time steps along with the policy. We use two critics, which has been shown to be sufficient for capturing epistemic uncertainty [14]. However, it is likely that the ensemble would be more effective with more value estimates, as demonstrated in [40].

## 6 Experiments

The key question we seek to address with our experiments is whether augmenting state-of-the-art off-policy actor-critic methods with TOP can increase their performance on challenging continuous-control benchmarks. We also test our assumption that the relative performance of optimistic and pessimistic strategies should vary across environments and across training regimes. We perform ablations to ascertain the relative contributions of different components of the framework to performance. Our code is available at `https://github.com/tedmoskovitz/TOP`.

**State-based control** To address our first question, we augmented TD3 [20] with TOP (TOP-TD3) and evaluated its performance on seven state-based continuous-control tasks from the MuJoCo framework [54] via OpenAI Gym [12]. As baselines, we also trained standard TD3 [20], SAC [21], OAC [14], as well as two ablations of TOP. The first, QR-TD3, is simply TD3 with distributional critics, and the second, non-distributional (ND) TOP-TD3, is our bandit framework applied to TD3 without distributional value estimation. TD3, SAC, and OAC use their default hyperparameter settings, with TOP and its ablations using the same settings as TD3. For tactical optimism, we set the possible $\beta$ values to be $\{-1, 0\}$, such that $\beta = -1$ corresponds to a pessimistic lower bound, and $\beta = 0$ corresponds to simply using the average of the critic. It's important to note that $\beta = 0$ is an optimistic setting, as the mean is biased towards optimism. We also tested the effects of different settings for $\beta$ (Appendix, Figure 6). Hyperparameters were kept constant across all environments. Further details can be found in Appendix B. We trained all algorithms for one million time steps and repeated each experiment with ten random seeds. To determine statistical significance, we used a two-sided t-test.

Table 1: Average reward over ten trials on Mujoco tasks, trained for 1M time steps. $\pm$ values denote one standard deviation across trials. Values within one standard deviation of the highest performance are listed in **bold**. $\star$ indicates that gains over base TD3 are statistically significant ($p < 0.05$).

| Task | TOP-TD3 | ND TOP-TD3 | QR-TD3 | TD3 | OAC | SAC |
|---|---|---|---|---|---|---|
| Humanoid | **5899$\pm$142$^\star$** | 5445 | 5003 | 5386 | 5349 | 5315 |
| HalfCheetah | **13144 $\pm$ 701$^\star$** | **12477** | 11170 | 9566 | 11723 | 10815 |
| Hopper | **3688 $\pm$ 33$^\star$** | 3458 | 3392 | 3390 | 2896 | 2237 |
| Walker2d | **5111 $\pm$ 220$^\star$** | 4832 | 4560 | 4412 | 4786 | 4984 |
| Ant | **6336 $\pm$ 181$^\star$** | 6096 | 5642 | 4242 | 4761 | 3421 |
| InvDoublePend | **9337 $\pm$ 20$^\star$** | **9330** | 9299 | 8582 | **9356** | **9348** |
| Reacher | **$-3.85 \pm 0.96$** | **$-3.91$** | **$-3.95$** | **$-4.22$** | **$-4.15$** | **$-4.14$** |

Our results, displayed in Figure 3 and Table 1, demonstrate that TOP-TD3 is able to outperform or match baselines across all environments, with state-of-the-art performance in the 1M time step regime for the challenging Humanoid task. In addition, we see that TOP-TD3 matches the best optimistic and pessimistic performance for HalfCheetah and Hopper in Fig. 2. Without access to raw scores for all environments we cannot make strong claims of statistical significance. However, it is worth noting that the mean minus one standard deviation of TOP-RAD outperforms the mean performance all baselines in five out of the seven environments considered.

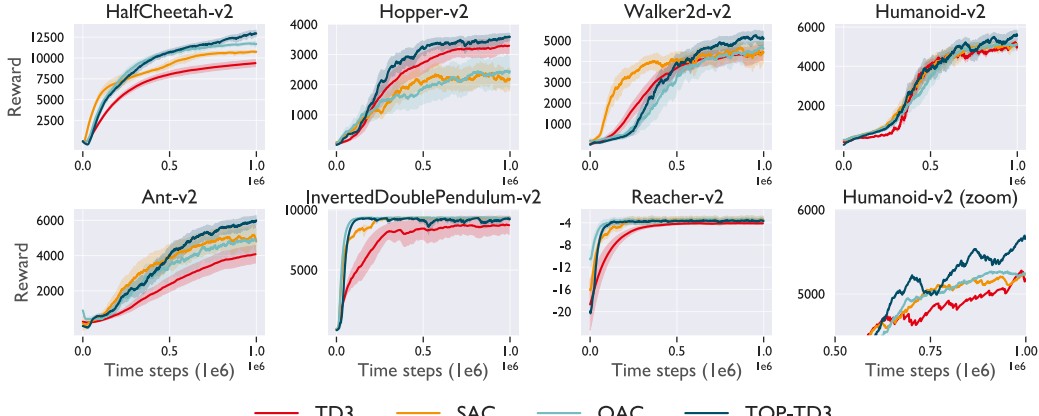

Figure 3: Reward curves for Mujoco tasks. The shaded region represents one half of a standard deviation over ten runs. Curves are uniformly smoothed. The lower right plot zooms in on the second half of the learning curve on Humanoid, omitting the shading for clarity.

Table 2: Final average reward over ten trials on DMControl tasks for 100k and 500k time steps. $\pm$ values denote one standard deviation across trials. Values within one standard deviation of the highest performance are listed in **bold**. $\star$ indicates that gains over base RAD are statistically significant ($p < 0.05$).

| Task (100k) | TOP-RAD | ND TOP-RAD | QR-RAD | RAD | DrQ | PI-SAC | CURL | PlaNet | Dreamer |
|---|---|---|---|---|---|---|---|---|---|
| Cheetah, Run | **674 $\pm$ 31$^\star$** | 600 | 546 | 499 | 344 | 460 | 299 | 307 | 235 |
| Finger, Spin | 873 $\pm$ 69 | 859 | 832 | 813 | 901 | **957** | 767 | 560 | 341 |
| Walker, Walk | **862 $\pm$ 43$^\star$** | 799 | 753 | 644 | 612 | 514 | 403 | 221 | 277 |
| Cartpole, Swing | **887 $\pm$ 13$^\star$** | 870 | 863 | 864 | 759 | 816 | 582 | 563 | 326 |
| Reacher, Easy | **991 $\pm$ 3$^\star$** | **991** | **992** | 772 | 601 | 758 | 538 | 82 | 314 |
| Cup, Catch | **970 $\pm$ 12$^\star$** | **959** | **962** | 950 | 913 | 933 | 769 | 710 | 246 |
| **Task (500k)** | **TOP-RAD** | **ND TOP-RAD** | **QR-RAD** | **RAD** | **DrQ** | **PI-SAC** | **CURL** | **PlaNet** | **Dreamer** |
| Cheetah, Run | **910 $\pm$ 4$^\star$** | 828 | 805 | 774 | 660 | 801 | 518 | 568 | 570 |
| Finger, Spin | 928 $\pm$ 74 | 920 | 911 | 907 | 938 | **957$^\star$** | **926** | 718 | 796 |
| Walker, Walk | **988 $\pm$ 4$^\star$** | 977 | 974 | 921 | 946 | 902 | 478 | 897 | |
| Cartpole, Swing | **890 $\pm$ 28$^\star$** | 873 | 866 | 858 | **868** | 816$^\star$ | 845 | 787 | 762 |
| Reacher, Easy | **993 $\pm$ 5$^\star$** | **994** | 956 | 930 | 942 | 950 | 929 | 588 | 793 |
| Cup, Catch | **972 $\pm$ 53$^\star$** | 970 | 971 | **970** | 963 | 933$^\star$ | 959$^\star$ | 939 | 879 |

**Pixel-based control** We next consider a suite of challenging pixel-based environments, to test the scalability of TOP to high-dimensional regimes. We introduce TOP-RAD, a new algorithm that

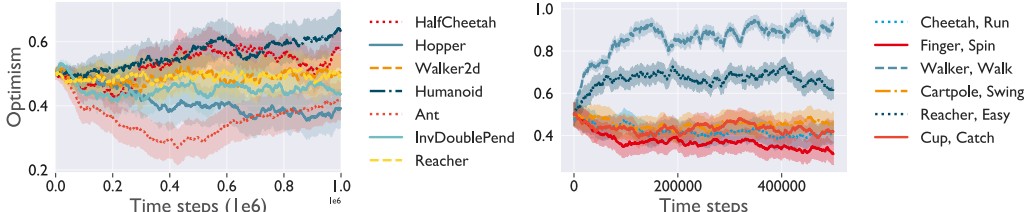

Figure 4: Mean optimism plotted across ten seeds. The shaded areas represent one half standard deviation.

dynamically switches between optimism and pessimism while using SAC with data augmentation (as in [31]). We evaluate TOP-RAD on both the 100k and 500k benchmarks on six tasks from the DeepMind (DM) Control Suite [52]. In addition to the original RAD, we also report performance from DrQ [58], PI-SAC [33], CURL [32], PlaNet [23] and Dreamer [22], representing state-of-the-art methods. All algorithms use their standard hyperparameter settings, with TOP using the same settings as in the state-based tasks, with no further tuning. We report results for both settings averaged over ten seeds (Table 2). We see that TOP-RAD sets a new state of the art in every task except one (Finger, Spin), and in that case there is still significant improvement compared to standard RAD. Note that this is a very simple method, requiring only the a few lines of change versus vanilla RAD—and yet the gains over the baseline method are sizeable.

**Does the efficacy of optimism vary across environments?** To provide insight into how TOP's degree of optimism changes across tasks and over the course of learning, we plotted the average arm choice made by the bandit algorithm over time for each environment in Figure 4. Optimistic choices were given a value of 1 and and pessimistic selections were assigned 0. A mean of 0.5 indicates that $\beta = 0$ (optimism) and $\beta = -1$ (pessimism) were equally likely. From the plot, we can see that in some environments (e.g., Humanoid and Walker, Walk), TOP learned to be more optimistic over time, while in others (e.g., Hopper and Finger, Spin), the agent became more pessimistic. Importantly, these changes were not always monotonic. On Ant, for example, TOP becomes steadily more pessimistic until around halfway through training, at which point it switches and grows more optimistic over time. The key question, then, is whether this flexibility contributes to improved performance.

To investigate this, we compared TOP to two baselines, a "Pessimistic" version in which $\beta = -1$ for every episode, and an "Optimistic" version in which $\beta$ is fixed to 0. If TOP is able to accurately gauge the degree of optimism that's effective for a given task, then it should match the best performing baseline in each task even if these vary. We tested this hypothesis

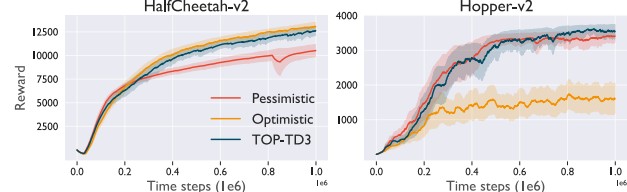

Figure 5: Mean performance of Pessimistic, Optimistic, and TOP across ten seeds. Shaded regions are one half standard deviation.

in the HalfCheetah and Hopper environments, and obtained the results shown in Figure 5. We see that TOP matches the Optimistic performance for HalfCheetah and the Pessimistic performance in Hopper. This aligns with Figure 4, where we see that TOP does indeed favor a more Optimistic strategy for HalfCheetah, with a more Pessimistic one for Hopper. This result can be seen as connected to the bandit regret guarantees referenced in Section 5.3, in which an adaptive algorithm is able to perform at least as well as the best *fixed* optimism choice in hindsight.

## 7 Conclusion

We demonstrated empirically that differing levels of optimism are useful across tasks and over the course of learning. As previous deep actor-critic algorithms rely on a fixed degree of optimism, we introduce TOP, which is able to dynamically adapt its value estimation strategy, accounting for both aleatoric and epistemic uncertainty to optimize performance. We then demonstrate that TOP is able to outperform state-of-the-art approaches on challenging continuous control tasks while appropriately modulating its degree of optimism.

One limitation of TOP is that the available settings for $\beta$ are pre-specified. It would be interesting to learn $\beta$, either through a meta-learning or Bayesian framework. Nevertheless, we believe that the bandit framework provides a useful, simple-to-implement template for adaptive optimism that could be easily be applied to other settings in RL. Other future avenues could involve adapting other parameters online, such as regularization [43], using natural gradient methods [38, 2], constructing the belief distribution from more than two critics, and learning a weighting over quantiles rather than simply taking the mean. This would induce a form of optimism and/or pessimism specifically with respect to aleatoric uncertainty and has connections to risk-sensitive RL, as described by [16, 36].

## Acknowledgements

We'd like to thank Rishabh Agarwal and Maneesh Sahani for their useful comments and suggestions on earlier drafts.

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
