# A Additional Experimental Results

The results for different settings of $\beta$ for TOP-TD3 on Hopper and HalfCheetah are presented in Figure 6.

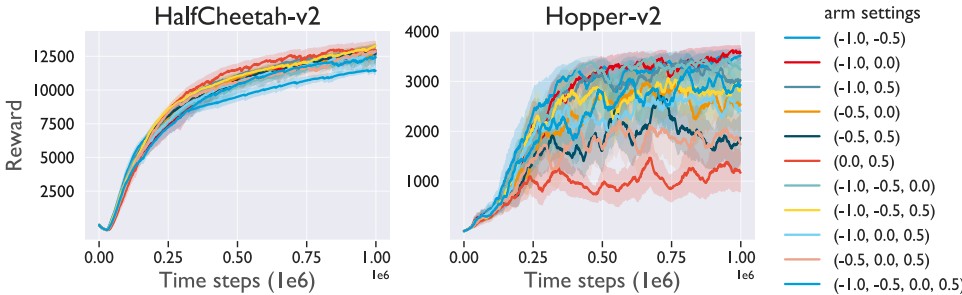

Figure 6: Results across 10 seeds for different sets of possible optimism settings. Shaded regions denote one half standard deviation.

Reward curves for TOP-RAD and RAD on pixel-based tasks from the DM Control Suite are shown in Figure 7.

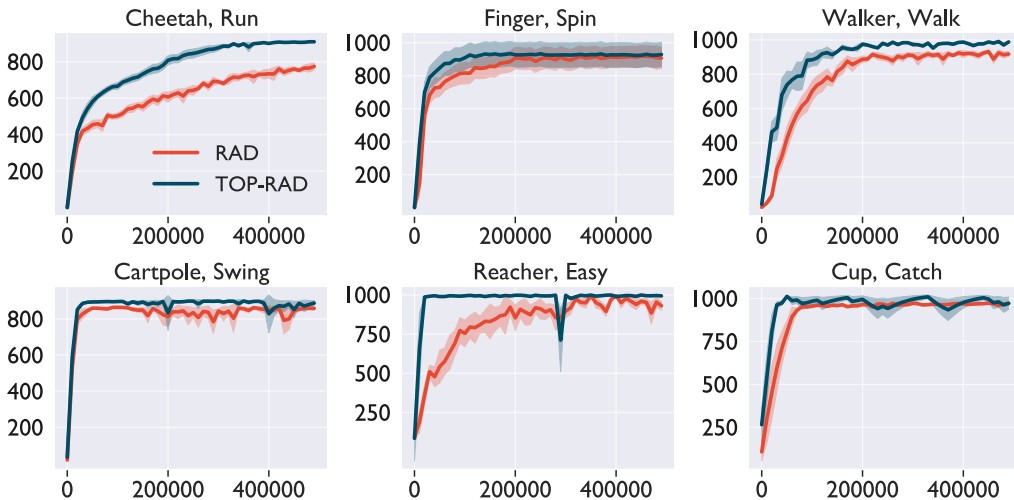

Figure 7: Results across 10 seeds for DM Control tasks. Shaded regions denote one half std.

# B Further Experimental Details

All experiments were run on an internal cluster containing a mixture of GeForce GTX 1080, GeForce 2080, and Quadro P5000 GPUs. Each individual run was performed on a single GPU and lasted between 3 and 18 hours, depending on the task and GPU model. The Mujoco OpenAI Gym tasks licensing information is given at `https://github.com/openai/gym/blob/master/LICENSE.md`, and the DM control tasks are licensed under Apache License 2.0.

Our baseline implementations for TD3 and SAC are the same as those from [7]. They can be found at `https://github.com/fiorenza2/TD3_PyTorch` and `https://github.com/fiorenza2/SAC_PyTorch`. We use the same base hyperparameters across all experiments, displayed in Table 3.

Table 3: Mujoco hyperparameters, used for all experiments.

| Hyperparameter | TOP | TD3 | SAC |
|---|---|---|---|
| Collection Steps | 1000 | 1000 | 1000 |
| Random Action Steps | 10000 | 10000 | 10000 |
| Network Hidden Layers | 256:256 | 256:256 | 256:256 |
| Learning Rate | $3 \times 10^{-4}$ | $3 \times 10^{-4}$ | $3 \times 10^{-4}$ |
| Optimizer | Adam | Adam | Adam |
| Replay Buffer Size | $1 \times 10^6$ | $1 \times 10^6$ | $1 \times 10^6$ |
| Action Limit | $[-1, 1]$ | $[-1, 1]$ | $[-1, 1]$ |
| Exponential Moving Avg. Parameters | $5 \times 10^{-3}$ | $5 \times 10^{-3}$ | $5 \times 10^{-3}$ |
| (Critic Update:Environment Step) Ratio | 1 | 1 | 1 |
| (Policy Update:Environment Step) Ratio | 2 | 2 | 1 |
| Has Target Policy? | Yes | Yes | No |
| Expected Entropy Target | N/A | N/A | $-\dim(\mathcal{A})$ |
| Policy Log-Variance Limits | N/A | N/A | $[-20, 2]$ |
| Target Policy $\sigma$ | 0.2 | 0.2 | N/A |
| Target Policy Clip Range | $[-0.5, 0.5]$ | $[-0.5, 0.5]$ | N/A |
| Rollout Policy $\sigma$ | 0.1 | 0.1 | N/A |
| Number of Quantiles | 50 | N/A | N/A |
| Huber parameter $\kappa$ | 1.0 | N/A | N/A |
| Bandit Learning Rate | 0.1 | N/A | N/A |
| $\beta$ Options | $\{-1, 0\}$ | N/A | N/A |

## C  Further Algorithm Details

The procedures for updating the critics and the actor for TOP-TD3 are described in detail in Algorithm 2 and Algorithm 3.

---

Algorithm 2: `UpdateCritics`

---

1: **Input:** Transitions $(s, a, r, s')_{n=1}^N$, optimism parameter $\beta$, policy parameters $\theta$, critic parameters $\phi_1$ and $\phi_2$.
2: Set smoothed target action (see (3))

$$\tilde{a} = \pi_{\theta'}(s') + \epsilon, \quad \epsilon \sim \text{clip}(\mathcal{N}(0, s^2), -c, c)$$

3: Compute quantiles $\bar{q}^{(k)}(s', \tilde{a})$ and $\sigma^{(k)}(s', \tilde{a})$ using (7).
4: Belief distribution: $\tilde{q}^{(k)} \leftarrow \bar{q}^{(k)} + \beta\sigma^{(k)}$
5: Target $y^{(k)} \leftarrow r + \gamma\tilde{q}^{(k)}$
6: Update critics using $\Delta\phi_i$ from (9).

---

---

Algorithm 3: `UpdateActor`

---

1: **Input:** Transitions $(s, a, r, s')_{n=1}^N$, optimism parameter $\beta$, critic parameters $\phi_1, \phi_2$, actor parameters $\theta$.
2: Compute quantiles $\bar{q}^{(k)}(s, a)$ and $\sigma^{(k)}(s, a)$ using (7).
3: Belief distributions: $\tilde{q}^{(k)} \leftarrow \bar{q}^{(k)} + \beta\sigma^{(k)}$
4: Compute values: $Q(s, a) \leftarrow K^{-1} \sum_{k=1}^K \tilde{q}^{(k)}$
5: Update $\theta$:

$$\Delta\theta \propto N^{-1} \sum \nabla_a Q(s, a)\big|_{a=\pi_\theta(s)} \nabla_\theta \pi_\theta(s).$$

---

Table 4: DM Control hyperparameters for RAD and TOP-RAD; TOP-specific settings are in purple.

| Hyperparameter | Value |
|---|---:|
| Augmentation | Crop - walker, walk; Translate - otherwise |
| Observation rendering | (100, 100) |
| Observation down/upsampling | (84, 84) (crop); (108, 108) (translate) |
| Replay buffer size | 100000 |
| Initial steps | 1000 |
| Stacked frames | 3 |
| Action repeat | 2 finger, spin; walker, walk |
| | 8 cartpole, swingup |
| | 4 otherwise |
| Hidden units (MLP) | 1024 |
| Evaluation episodes | 10 |
| Optimizer | Adam |
| $(\beta_1, \beta_2) \rightarrow (f_\theta, \pi_\psi, Q_\phi)$ | $(0.9, 0.999$ |
| $(\beta_1, \beta_2) \rightarrow (\alpha)$ | $(0.5, 0.999$ |
| Learning rate $(f_\theta, \pi_\psi, Q_\phi)$ | 2e-4 cheetah, run |
| | 1e-3 otherwise |
| Learning rate $(\alpha)$ | 1e-4 |
| Batch size | 128 |
| $Q$ function EMA $\tau$ | 0.01 |
| Critic target update freq | 2 |
| Convolutional layers | 4 |
| Number of filters | 32 |
| Nonlinearity | ReLu |
| Encoder EMA $\tau$ | 0.05 |
| Latent dimension | 50 |
| Discount $\gamma$ | 0.99 |
| Initial Temperature | 0.1 |
| Number of Quantiles | 50 |
| Huber parameter $\kappa$ | 1.0 |
| Bandit Learning Rate | 0.1 |
| $\beta$ Options | $\{-1, 0\}$ |

## D  Connection to Model Selection

In order to enable adaptation, we make use of an approach inspired by recent results in the model selection for contextual bandits literature. As opposed to the traditional setting of Multi-Armed Bandit problems, the "arm" choices in the model selection setting are not stationary arms, but learning algorithms. The objective is to choose in an online manner, the best algorithm for the task at hand. The setting of model selection for contextual bandits is a much more challenging setting than selecting among rewards generated from a set of arms with fixed means. Algorithms such as CORRAL [1, 44] or regret balancing [42] can be used to select among a collection of bandit algorithms designed to solve a particular bandit instance, while guaranteeing to incur a regret that scales with the best choice among them. Unfortunately, most of these techniques, perhaps as a result of their recent nature, have not been used in real deep learning systems and particularly not in deep RL.

While it may be impossible to show a precise theoretical result for our setting due to the function approximation regime we are working in, we do note that our approach is based on a framework that under the right settings can provide a meaningful regret bound. In figure 5 we show that our approach is able to adapt and compete against the best *fixed* optimistic choice in hindsight. These are precisely the types of guarantees that can be found in theoretical model selection works such as [1, 44, 42]. What is more, beyond being able to compete against the best fixed choice, this flexibility may result in the algorithm *outperforming* any of these. In figure 5, Ant-v2 we show this to be the case.

# E   Proofs

*Proof of Proposition 1.* Let $q_{Z^\pi}$ be the quantile function of $Z^\pi(s, a)$ knowing $\epsilon$ and $\sigma$ and $q_{\bar{Z}}$ be the quantile function of $\bar{Z}$. Since $\epsilon$ and $\sigma$ are known, the quantile $q_{Z^\pi}$ is given by:

$$q_{Z^\pi}(u) = q_{\bar{Z}}(u) + \epsilon\sigma(s, a).$$

Therefore, recalling that $\epsilon$ has 0 means and is independent from $\sigma$, it follows that

$$q_{\bar{Z}}(u) = \mathbb{E}_\epsilon\left[q_{Z^\pi}(u)\right]$$

The second identity follows directly by definition of $Z^\pi(s, a)$:

$$Z^\pi(s, a) = \bar{Z}(s, a) + \epsilon\sigma(s, a).$$

$\square$