# OpenReview forum: "Tactical Optimism and Pessimism for Deep Reinforcement Learning"
_NeurIPS.cc/2021/Conference — NeurIPS 2021 Poster_

### Official Review · Reviewer_fVYV · 2021-07-13

**Rating:** 6
**Confidence:** 4

**Summary:**

**Summary and contributions**

This paper studies an interesting contradiction in the reinforcement learning literature, which suggests that both optimistic and pessimistic attitudes toward exploration can lead to high performance. The paper argues that no single attitude applies equally-well to all domains and to all stages of learning; for example, it can sometimes be beneficial to explore pessimistically in the beginning and then slowly change to adopt an optimistic attitude after some amount of experience.

 To resolve the apparent contradiction, the paper proposes the algorithm called TOP. TOP estimates epistemic uncertainty and uses it to bias sample estimates of a distributional reinforcement learning algorithm. The magnitude of bias is adapted online using one of the EXP bandit algorithms, with meta-like feedback in the form of differential episodic returns. Thus, the system will choose the exploration strategy to improve the utility of its learning process.

The paper applies TOP to algorithms based on the actor-critic architecture, and it evaluates the algorithm extensively in continuous control domains, showing positive results to suggest the extra adaptation is beneficial.

**Main Review:**

**General impressions**

I quite like this paper. I think that it contains many interesting ideas, and it presents solid empirical evidence that support its main claims. But with that said, it currently feels like a stretch to call the paper complete. I have a few issues with the writing and presentation, and there are some technical details that I’d like to clarify with the authors before committing to accept the paper.

**Further details**

In Section 1, the paper does a good job of motivating its research problem by presenting an apparent contradiction. But it follows this up with a vague hypothesis (Lines 36-37). This can be fixed by making the statement more specific and using terms that have been previously introduced. Remove loose language such as “function of the environment”, and “overall context in which a learner is embedded”. The content of a hypothesis needs to be specific and unambiguous enough to be falsifiable.  These comments also apply to the claim on line 44.

In general the language needs to be improved.
* “estimates the optimal approach” (Line 48). This algorithm can’t claim to be optimal, but it can claim to be good or useful.
* “Learning-theoretic” (Line 43): This is a misuse of the term.
* I think it’s wrong to say pessimism and optimism are in balance, because choosing one doesn’t come at the expense of the other.
* “A Bayesian representation of aleatoric uncertainty” is a contradiction in terms. As more information is collected, a Bayesian representation of uncertainty will eventually vanish. But aleatoric uncertainty will persist despite the amount of experience the system has seen.
* “Degree of optimism” is vague. The system has an attitude toward exploration, which can be pessimistic or optimistic. Choosing an attitude gives the system an exploration strategy.
* The main contributions need to be more clearly stated, without ambiguous language.

The way epistemic uncertainty is discussed and estimated is confusing (Lines 189--196). It took me several passes to realize that epistemic uncertainty referred to the estimation error of the return distribution’s quantile. The notation and text do not make this clear.

The random return does not have to be modeled as a Gaussian. One can decompose a random variable into its mean and zero-mean residual. I believe this is what is done in Equation 8. Saying this to start with and removing the language around Gaussians will simplify the exposition.

Proposition 1 seems obvious and potentially not helpful. Unless the authors can justify its purpose, I suggest it be removed.

The notation is a bit of a disaster. Random variables and distributions are denoted the same way. Equation 5 uses a random variable as the subscript of the quantile function, but Equation 8 uses the belief distribution as the subscript. Equation 4 should reflect equality in distribution. Indices are used without defining their range. I strongly recommend taking another editing pass to simplify and clarify notation.

How important are having multiple estimates of the quantile function for epistemic uncertainty? Right now you have two.

Can you justify the choice for the baselines and describe what the differences between the proposed algorithm and these represent?

**Other feedback / questions**

* Why is some text of your algorithm red?
* Aleatoric and Epistemic uncertainty are introduced as terms without definition. It is better to describe the idea of a concept before the term, if it’s not going to be defined.
* How was significance determined in Table 1?


**Time Spent Reviewing:**

7

---

> ### Author Response · Authors · 2021-08-10
> **Authors' Response**
>
> Thank you very much for your thoughtful review and detailed comments. We’re glad that you liked the paper! We hope we can address your concerns.
>
> - First, thank you for your many helpful points regarding imprecise language and notation--we will certainly modify the paper accordingly.
>
> - Notation: We apologize for this, we believe the confusion comes from calling $Z(s,a)$ a ’return  distribution’  and then treating it as a random variable in equation 4. We will clarify in the text that $Z(s,a)$ is a random variable representing the return distribution and that all distributions we consider are represented through a random variable. This convention was also used in Dabney et al. (2018) and  should resolve the confusion about eq 4, 5 and 8 as they will always deal with a random variable $Z$ and never directly with a distribution.
>
> - We believe that having multiple value estimates is crucial for modeling epistemic uncertainty, an idea supported in recent literature [e.g., Ref. 38]. While having two estimates seems to be sufficient to improve performance in practice, adding more value heads would only improve this representation, albeit at increased computational/memory cost, and is an interesting future direction.
>
> - The most important baselines for state-based and pixel-based control are TD3 and RAD, respectively, as we modify these approaches with TOP. The other baselines are largely included because they constitute the “state-of-the-art” for their respective domains, and thus we believe that some point of comparison for our method is useful.
>
> - As we say in lines 263-4, the colored text highlights differences from standard TD3.
>
> - Significance was determined using a two-sided t-test. This should have been included, and we’ll certainly add this detail to the text.

---

> > ### Comment · Reviewer_fVYV · 2021-08-23
> > **Reply**
> >
> > Thank you for your response. I don't have much to add to my review, just a few comments and questions.
> >
> > Re: notation of the random return. I think you understood my point here, that calling the random return a distribution is confusing and technically incorrect. This was an unfortunate convention used in the first few distributional RL papers that I hope subsequent work can correct.
> >
> > Re: multiple value estimates. I think more should be said here to justify the use of two value heads. Although it appears sufficient to improve performance, it amounts to using two samples in an ensemble, whose estimate of uncertainty is questionable. A reader will wonder whether the performance changes are due to the uncertainty being represented or something else, such as a hidden regularization effect. Ideally the paper would contain an additional experiment to investigate this, either by ruling out regularization or showing the uncertainty estimate is clearly important.
> >
> > Re: EXP parameters. In my experience the EXP algorithms are quite sensitive to their parameterization. Could you comment on sensitivity of  your results to these parameters?

---

> > > ### Author Response · Authors · 2021-08-30
> > > **Response**
> > >
> > > Thank you very much for your insightful comments and questions.
> > >
> > > Re: notation of the random return. Yes, absolutely—this is a good point. We initially erred on the side of convention, but agree that the proper description is preferable.
> > >
> > > Re: multiple value estimates. Thank you, this is also an interesting point. The multiple value heads can be thought of representing epistemic uncertainty in a similar way to Bootstrap DQN [1], although we are not performing approximate Thompson sampling. The interplay between these representations and implicit regularization is very interesting and tricky to disentangle, and there’s recent evidence that implicit regularization can actually be harmful when learning deep value estimates (see the recent “Value-Based Deep Reinforcement Learning Requires Explicit Regularization” by Kumar et al. as an example). Using only a single critic harms performance in both the TOP-based algorithms and the baselines. We think this is an interesting topic that merits further investigation.
> > >
> > > Re: EXP parameters. We simply used the default parameters from a previous implementation of EXP3, but we agree it would be interesting to probe the robustness of TOP to different settings. We do note that we did experiment with varying numbers of arms (Fig. 6 in the appendix).
> > >
> > > Thank you once again!
> > >
> > > [1] https://arxiv.org/abs/1602.04621

---

### Official Review · Reviewer_Xse6 · 2021-07-13

**Rating:** 7
**Confidence:** 4

**Summary:**

The paper introduces a bandit algorithm to adaptively change optimisim/pesimism in the actor critic algorithms


**Limitations And Societal Impact:**

Sufficiently addressed

**Main Review:**

In many ways, it looks like the paper builds on top of “K. Ciosek, Q. Vuong, R. Loftin, and K. Hofmann, Better exploration with optimistic actor-critic, In Advances in Neural Information Processing Systems”. While that work considers only optimistic setup (where beta is positive), this work allows for both positive and negative beta.
The idea is to use (discretized) beta in a multi-arm bandit problem to adapt the optimism/pesimism per task (and also during training)

The paper is well written and clear. Previous work seems to be well cited. The motivation is well explained, and further supported by the experiments in the last (sub)section.
Authors also include quite a few Mujoco tasks to evaluate the method and compare it to previous ones. The evaluation is also done very well,

I like the paper, and I see only two negative points here. First, the resulting comparison to previous methods is not that amazing - it mostly does better (to be fair, that is usually the case for many algorithms, it’s very rare to substantially do better in all domains). Second, the work is very much incremental to prior methods.





**Time Spent Reviewing:**

6

---

> ### Author Response · Authors · 2021-08-10
> **Authors' Response**
>
> Thank you for your thoughtful review--we’re glad you liked the paper! To briefly address your concern regarding incremental improvement--we absolutely agree that the changes we make are simple ones. However, we believe that this is in fact a strength of our approach. TOP is a simple modification that improves the state-of-the-art across domains, and which is agnostic to the details of the algorithm to which it is applied.

---

### Official Review · Reviewer_o4fR · 2021-07-15

**Rating:** 6
**Confidence:** 4

**Summary:**

This paper is motivated by the two (seemingly contradicting) views of learning efficiently in the RL setup, that of pessimism and optimism. It touches upon why pessimism has been useful in computing better value estimates when deploying function approximation. It then argues that pessimism also leads to reduced exploration and contradicts the optimism favouring view predominant in RL theory. Finally, it claims that the right levels of pessimism/optimism are problem dependent and thus proposes adding a bandit algorithm to adaptively choose either optimism or pessimism for individual problems. This algorithm (called TOP) is tested on both state-based and pixel-based MuJoCo benchmarks. There is a decent amount of performance improvement seen in both cases when comparing against popular baseline methods.

**Ethical Concerns:**

I do not think there any explicit ethical concerns raised by this work.

**Limitations And Societal Impact:**

Suggestions: I believe making contributions in the first category (as described above) would be really valuable. In particular, if we can have an experiment with a simple task such as Mountain Car, and show that the amount of exploration directly varies with the tradeoff in optimism/pessimism. The effect pessimism and optimism have on the exploration seems to be the key to better performance (maybe there is something else as well? maybe they directly affect the representations learnt?) and this does not come forth in the experiments with more complex domains. Furthermore, going deeper into why/how the representations get better when we apply the proposed adaptive strategy can give even more insights.


I do not think there any explicit negative societal impacts of this work.

**Main Review:**

Originality: The motivation of the work is solid in my opinion. The main question the authors ask is indeed important and I do not know of any other works that empirically test for the trade-off between pessimism and optimism (though I do not follow this exact topic in too much depth). In terms of the solution proposed, I do not rate it too highly in terms of originality.

Quality: The overall quality of the work is quite high. The experiments are well executed and observations are nicely reported.

Clarity: The paper reads quite clearly overall. The flow of content is pretty nice as well.

Significance: Since the main contribution of the paper is empirical, there could be two ways to evaluate the significance in my opinion: 1) how much insights do the experiments shed on the trade-off between pessimism and optimism depending on the problem in hand, and 2) how much empirical gains can one get from the proposed solution + how simple would the method be to implement. I strongly feel that this paper makes contributions in the second category. In that regard, I do not completely get why the distributional framework is necessary here. It seems that the pessimism/optimism trade-off can be modelled without bringing in the distributional framework, as is done in the ND TOP-TD3 method. Therefore, to me it seems as if it was added just to boost the performance further and add more content for the paper. That is fine in general but I would argue that it then hampers the simplicity component, and does not highlight as well how much difference the bandit algorithm can provide in and of itself.


Questions:

1. Is there any special reason to choose to work with TD3 instead of SAC? I believe both use the pessimistic form for updating the value functions.

2. The authors say that the code is attached but I could not find any. Can the authors please share the code (with a read-me)?

3. Was there a similar bifurcation done for the pixel-based case like in the state-based case? i.e. ND TOP-TD3 and QR-TD3 for the pixel-based method?

4. I would really like to see error statistics for all methods, instead of just for the TOP method in the tables.

5. Based on the above comments, I am inclined to look at the ND TOP-TD3 column as the main results. Is that a fair way to interpret the results? Can the authors explain more about the inclusion of the distributional framework in this respect?



**Time Spent Reviewing:**

5 hrs

---

> ### Author Response · Authors · 2021-08-10
> **Authors' Response**
>
> Thank you very much for your thorough and detailed review. We’re very glad that you find the problem we address to be important, the work to be of high quality, and the paper itself clearly written and well-organized. We are also glad that you note the significance of the performance increase with respect to the simplicity of the method. We hope we can address your concerns:
>
> Significance: As you mentioned, the experiments illustrate  how much gain one can expect from the method as well as its applicability to a wide range of settings (point (2) in your review). We also show that the tradeoff between optimism and pessimism varies not only across tasks but also within each task and across time (see Figure 4 and 5). To our knowledge this was not known. But we agree that precisely characterizing the degree of optimism needed for a given problem is a challenging and interesting future work direction.
>
>
> To address your specific questions:
> There isn’t a specific reason to use TD3 over SAC, we believe TOP would work equally well applied to either algorithm, as they are closely related (Ref. [6] in the paper). In fact, RAD itself is built on top of SAC, so TOP-RAD can also be seen as building on it.
> Apologies! We intended to attach the code. Here’s a link to an anonymized repository: https://anonymous.4open.science/r/TOP-16C3/README.md
> Yes, we also have results for ablated versions of TOP-RAD. We list them below in a table, and we’ll add them to the paper.
>
> | task (100k) | ND TOP-RAD   | QR-RAD       |
> | ----------- | ------------ | ------------ |
> | cheetah     | 600 $\pm$ 29 | 546 $\pm$ 36 |
> | finger      | 859 $\pm$ 24 | 832 $\pm$ 15 |
> | walker      | 799 $\pm$ 48 | 753 $\pm$ 29 |
> | cartpole    | 870 $\pm$ 18 | 863 $\pm$ 5  |
> | reacher     | 991 $\pm$ 4  | 992 $\pm$ 6  |
> | cup         | 959 $\pm$ 2  | 962 $\pm$ 6  |
>
> | task (500k) | ND TOP-RAD   | QR-RAD       |
> | ----------- | ------------ | ------------ |
> | cheetah     | 828 $\pm$ 22 | 805 $\pm$ 5  |
> | finger      | 920 $\pm$ 60 | 911 $\pm$ 38 |
> | walker      | 977 $\pm$ 5  | 974 $\pm$ 4  |
> | cartpole    | 873 $\pm$ 10 | 866 $\pm$ 8  |
> | reacher     | 994 $\pm$ 2  | 956 $\pm$ 3  |
> | cup         | 970 $\pm$ 5  | 971 $\pm$ 6  |
>
> For the state-based tasks we were able to run all of the baseline methods ourselves, but for computational reasons we had to rely on previously reported results for some pixel-based baselines. For the state-based tasks, we’ve shown the error shading in Figure 3, and we can add the information to Table 1, space permitting. We will reach out to the authors of the remaining papers to ask for error statistics.
> We consider the TOP results to be the main results. A core part of our approach is to leverage estimates of both epistemic and aleatoric uncertainty in order to intelligently adjust the agent’s degree of optimism. The former is captured via multiple value estimates and the latter via distributional critics. As described in Section 5.1 (in particular, eq. 4, Proposition 1, and eq. 7), however, these quantities are not independent, and are combined to decompose the return distribution into its mean and standard deviation. Eq. 7 shows how an approximate return distribution is constructed by taking the *quantile-wise* average and standard deviation of the critic distributions. While an analogous computational process can be carried out with scalar critics, the result does not contain the same (aleatoric) information as the quantile-based construction. With this being said, we believe that the strong performance of ND-TOP alone is also evidence in favor of our framework.
>
>
> We hope we’ve adequately addressed your questions and hope you will consider raising your score.  We are also happy to discuss any remaining concerns further!

---

> > ### Comment · Reviewer_o4fR · 2021-08-22
> > **Thanks!**
> >
> > Thank you for the detailed reply and for running the additional experiments on RAD. This has definitely addressed my questions and so I am updating the score accordingly.

---

### Official Review · Reviewer_b4Gb · 2021-07-16

**Rating:** 9
**Confidence:** 5

**Summary:**

The paper considers the problem of approximate value estimation in continuous-action actor-critic architectures, raising the argument that the overestimation bias is not always (in all environments) bad. In particular, they accord to the *optimism in the face of uncertainty* principle to argue that overestimation helps in some domains due to promoting exploration, while in others the negative impacts of such bias outweigh the exploratory benefits (which is why pessimistic approaches such as TD3 have improved performance in many domains). Based on this argument the direction of the paper is a natural one: create a method that adapts the agent's degree of optimism in a single training lifetime. The authors propose such an approach where (at least) two action-value critics are trained, based on which a measure of *epistemic uncertainty* can be obtained. Based on this epistemic uncertainty, the agent's value estimation can be adjusted to be more optimistic or pessimistic. To isolate this measure from *aleatoric uncertainty*, they use distributional critics. There is only a single scalar parameter, $\beta$, that determines the degree of optimism, and the proposed method applies a multi-armed bandit formulation across episodes to optimize this parameter (over discretized values of the parameter in the range of $[-1, 0]$).

**Main Review:**

I salute the authors for this fine paper. The paper packs a lot of information but so concisely and clearly.

The paper presents a fresh take on optimism/pessimism in approximate value estimation. Most successful methods in continuous control address the overestimation of values with pessimism, whereas Ref. [13] proposes an optimistic actor-critic approach that achieves better performance. As such, the present work addresses a natural gap by maintaining the best of both worlds adaptively in the context of the environment.

The paper is written beautifully. The ideas are clearly presented. The extensions needed to implement the method are minimalistic. I like the fact that the paper doesn't try to use meta-learning, or other more complicated settings, to showcase the base ideas. This makes the paper easy to follow, without confounding effects that would make validating the fundamental ideas difficult.

The arguments are well motivated with experiments in Hopper and HalfCheetah domains (see Fig. 5). The experiments are thorough (the number of seeds is sufficient, baseline set is reasonable and up-to-date), showing that improvements across all domains can be obtained with a single algorithm. Also, one interesting finding is that (Fig. 4 left) the degree of optimism need not be constant or monotonic during the training course; this further illustrates the utility of using adaptive optimism during training.

**Minor:**
- L116: This is really Q-learning rather than SARSA because $\pi(s_{t+1})$ is not generally the action that the agent has executed in the environment; it's critical to off-policy learning in DPG-style methods to use a Q-learning-style Bellman update on the critic.
- Why focus only on continuous-action algorithms? Doesn't the same ideas hold in discrete-action methods? E.g. Q-learning vs Double Q-learning?
- Why not use $\beta > 0$ in experiments?

**Time Spent Reviewing:**

8

---

> ### Author Response · Authors · 2021-08-10
> **Authors' Response**
>
> Thank you very much for your insightful review and comments. We are glad you enjoyed the paper! To address your minor points:
>
> Q-learning rather than SARSA:Thank you, we’ll clarify this in the text.
>
> Continuous vs Discrete:  It’s certainly true that the same principle could be applied in tasks with discrete action spaces--we focus on continuous control here only because we originally spotted the gap between purely optimistic and pessimistic approaches in this setting. Applying TOP to discrete environments would be an interesting opportunity for follow-up work.
>
> Choice of $\beta$: Arm settings with $\beta > 0$ are also effective--see Fig. 6 in the appendix for results with different arm settings. We use $\beta \in \{-1, 0\}$ in the main results simply because that’s what we tried first.

---

> > ### Comment · Reviewer_b4Gb · 2021-08-10
> > **Thanks for your comments**
> >
> > Thanks for clarifying the points regarding the choices of $\beta$ and discrete action spaces. It could help to mention these briefly in the main text as well in my view.

---

> > > ### Author Response · Authors · 2021-08-18
> > > **Response**
> > >
> > > We'll certainly add those clarifications to the main text. Thank you once again!

---

### Decision · Program_Chairs · 2021-09-27

**Decision:**

Accept (Poster)

**Comment:**

This is a solid piece of work and the reviewers agreed in the end. The paper is well written, it unifies two views on exploration, the experiments are well done, and the authors did a good job both (a) responding to the reviews and (b) improving on the previous submission. There were some concerns about the working being incremental, but consensus emerged that the work exhibited a high degree of correctness (which is critical and not as common as should be) that the paper provides useful insights and foundations of future work. Another reviewer was generally favourable after discussion but did not update their score to reflect that. Given the relevance and difficulty of efficient exploration this work will make a welcome addition to the neurips program.

There were some important missing details and notational issues flagged by one reviewer. I am confident these can be addressed easily based on the author response.